# Fine-Grained Matching with Multi-Perspective Similarity Modeling for Cross-Modal Retrieval

**Xiumin Xie**         **Chuanwen Hou**         **Zhixin Li** *

Guangxi Key Lab of Multi-source Information Mining and Security, Guangxi Normal University, Guilin 541004, China

## Abstract

Cross-modal retrieval relies on learning inter-modal correspondences. Most existing approaches focus on learning global or local correspondence and fail to explore fine-grained multi-level alignments. Moreover, it remains to be investigated how to infer more accurate similarity scores. In this paper, we propose a novel fine-grained matching with Multi-Perspective Similarity Modeling (MPSM) Network for cross-modal retrieval. Specifically, the Knowledge Graph Iterative Dissemination (KGID) module is designed to iteratively broadcast global semantic knowledge, enabling domain information to be integrated and relevant nodes to be associated, resulting in fine-grained modality representations. Subsequently, vector-based similarity representations are learned from multiple perspectives to model multi-level alignments comprehensively. The Relation Graph Reconstruction (SRGR) module is further developed to enhance cross-modal correspondence by constructing similarity relation graphs and adaptively reconstructing them. Extensive experiments on the Flickr30K and MSCOCO datasets validate that our model significantly outperforms several state-of-the-art baselines.

## 1 INTRODUCTION

Cross-modal retrieval refers to retrieving the most relevant text (image) by utilizing the image (text) as a query. Its core is to detect the potential correlation between different modalities and then measure cross-modal similarity to achieve relatively accurate matching [Hou et al., 2021].

Existing methods mainly learn global or local alignment between image and text for retrieval. The global alignment

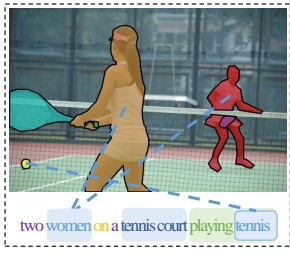
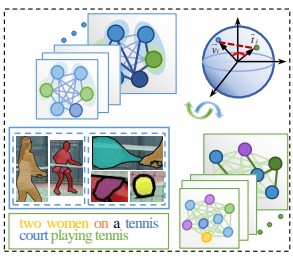

(a) Coarse-grained alignment         (b) Fine-grained alignment

Figure 1: Illustration of coarse and fine-grained alignment.

learning methods [Hardoon et al., 2004, Karpathy and Li, 2015, Zheng et al., 2020] aim to discover correspondences between entire image and text, but fail to investigate fine-grained correspondence between regions and words. As a result, local alignment learning methods [Niu et al., 2017, Chen et al., 2019, Chen and Luo, 2020] are presented that are used to model the region-word correspondence. However, global or local alignment is one-sided. Therefore, some researchers jointly model global and local correspondences to obtain multi-level alignments. The multi-level alignment methods [Huang et al., 2018a, Li et al., 2020, Nguyen et al., 2021, Yuan et al., 2021] can provide more complimentary as well as comprehensive semantic cues, thus improving performance. Furthermore, fine-grained alignment should account for both inter-modal and intra-modal correlations.

More importantly, similarity representation and learning are key to cross-modal matching. Most approaches [Chen et al., 2020, Wang et al., 2020, Li et al., 2019] compute scalar-based cosine distances to reflect the cross-modal similarity. However, this merely yields a constant scalar value and fails to adaptively refine the visual-semantic correspondence. Recently, some novel methods [Kuang et al., 2019, Diao et al., 2021] have achieved excellent results by representing similarity with vectors rather than scalars. These methods, however, use a single form of similarity representation and fail to learn cross-modal similarity in a fine-grained manner.

---

*Zhixin Li is the corresponding author.

*Accepted for the 38th Conference on Uncertainty in Artificial Intelligence* (UAI 2022).

In summary, there are several challenges for fine-grained image-text retrieval. To begin, it needs to consider both global and local alignment, as they facilitate the interaction between the "object" and the "global context", as shown in Figure 1, where the region-global interaction "women on a tennis court". Second, existing methods fail to explore fine-grained intra-modal correlation which can provide richer semantic information. As illustrated in Figure 1(b), the interaction between the "racket", the "ball", and the women's "arm" regions corresponds to the phrase "playing tennis". Significantly, the representation of similarity and its learning should be considered from multiple perspectives. For example, "A woman is hitting a tennis." is semantically related to the image in Figure 1(vectors are in the same direction), yet they are mismatched (there are numerical differences).

Motivated by these, we propose a novel fine-grained matching with Multi-Perspective Similarity Modeling (MPSM) Network for cross-modal retrieval. Specifically, we first construct visual and textsual semantic knowledge graphs. Then, we introduce the Knowledge Graph Iterative Dissemination (KGID) module that learns fine-grained intra-modal correlations and modal representations by iteratively propagating the knowledge. Subsequently, we learn vector-based similarity representations from multiple perspectives separately to model multi-level alignment. The proposed similarity representations are learnable and can comprehensively explore image-text correspondences. Furthermore, we designed the Similarity-Relation Graph Reconstruction (SRGR) module to achieve more accurate matching by constructing similarity relation graphs and adaptively reconstructing similar relations. Our main contributions are summarized below:

- We propose a KGID module that integrates domain information of nodes and iteratively propagates semantic information to neighboring nodes to capture fine-grained local and global representations.

- We learn vector-based similarity from multiple perspectives, which allows for more comprehensive learning of multi-level correspondences. To our knowledge, this is the first work to considers both distance and direction similarity for similarity representation and learning.

- We design a SRGR module in which the similarity relational graphs are constructed and reconstructed adaptively to achieve information interaction between multi-level alignments, filter interference and enhance similarity, thus improve matching accuracy.

## 2 RELATED WORK

### 2.1 IMAGE-TEXT RETRIEVAL

Existing methods can be roughly split into global alignment, local alignment and multi-level alignment learning methods.

The global alignment learning methods seek to learn correspondences between the entire image and text. Frome et al. [2013] were the first to map the full image and text into a common space for semantic alignment. Some approaches are inspired by generative adversarial network (GAN)[Goodfellow et al., 2014], Wang et al. [2017] employs GAN to produce features. There were also methods focusing on optimization, Faghri et al. [2018] presented an optimization scheme that increases the distance between samples and hard samples. Furthermore, Wang et al. [2018] emphasized the need of maintaining both inter-modal and intra-modal correspondence. Nevertheless, the above mothods neglected fine-grained semantic associations between regions and words, as well as intra-modal associations.

The local alignment learning methods explore region-word correspondence to acquire more accurate similarities. Karpathy et al. [2014] made the first attempt by combining the alignment of related region-word pairings. Lee et al. [2018] used an attention mechanism to align each region with all words, verifying the efficiency of region-word alignment. Many of the following works were based on [Lee et al., 2018]. For example, Wang et al. [2019b] followed [Lee et al., 2018] to model the region-word relation. Several motheds focused on both inter-modal and intra-modal relations, such as [Liu et al., 2019, Zhang et al., 2020, Wei et al., 2020]. However, these mothods failed to comprehensively explore fine-grained visual-semantic similarity. Unlike, we dynamically explore intra-modal correlations and model multi-level correspondences for more complete alignment.

Recently, researchers are increasingly exploring both global correspondence and local correspondence to measure cross-modal similarity comprehensively[Qi et al., 2018, Ma et al., 2019, Wen et al., 2020]. Some approaches first tried [Qi et al., 2018, Ma et al., 2019] to tackle the image-text matching in a multi-pathway, computing the global and local similarities, and combining them into the final similarity. However, these approaches ignore that a word or a region may have different semantics in different global contexts, while global contextual information can be used as a clue for semantically similar samples [Wei et al., 2021, Xian et al., 2022]. Based on this, Ji et al. [2021] implemented local-to-local, global-to-local, and global-to-global cross-modal alignments in turn. Further, Diao et al. [2021] computed similarity representations for all local and global representation pairs simultaneously. However, existing multi-level alignment methods were insufficient for the learning of similarity, capturing only limited information.

### 2.2 SIMILARITY REPRESENTATION LEARNING

The core of cross-modal retrieval is the learning of similarity between different modalities. As for global alignment methods, some [Faghri et al., 2018, Wang et al., 2016, Li et al., 2021] explored similarity by computing the cosine distance.

Another branch [Vendrov et al., 2015, Huang et al., 2018b, Gu et al., 2018] introduced ordered representations. As for local alignment, most methods [Liu et al., 2019, Wang et al., 2019b] computed scalar-based cosine distance to reflect region-word similarity. Furthermore, most multi-level alignment methods [Qi et al., 2018, Ma et al., 2019] modeled local and global alignments separately by using scalar-based cosine distance and combined them to reflect the similarity.

The above approaches' similarity representations are scalar values that cannot learn fine-grained visual-semantic correlations adaptively. Recently, some innovative approaches [Diao et al., 2021] to similarity representation and learning have been developed, Diao et al. [2021] learned vector-based similarity representation to explore multi-level alignment, and achieved some improvement. However, for similarity, it appears to be one-sided and fails to learn the correlation between vectors in a thorough manner. Differently, we learns vector-based similarities from multiple perspectives to model multilevel alignment more comprehensively.

## 2.3 THE DIFFERENCE WITH OTHER METHOD

Compared to the GSMN [Liu et al., 2020], which also captures semantic information, while it lacks the mining of fine-grained intra-modal interactions. Instead of conducting basic matching, we consider integrating and spreading semantic knowledge among nodes. It allows for the dynamic mining of intra-modal correlations as well as the capturing of semantically rich features. Besides, comparison to SGRAF [Diao et al., 2021], which also utilizes graph reasoning. One of the key differences is that we not only mine the rich semantic information within the modality but also adaptively associates relevant nodes. It thus enables semantically rich intra-modal correlations to be included in inter-modal similarity learning and inference, which SGRAF [Diao et al., 2021] does not make possible.

Another aspect, we cleverly model multi-level alignments and perform similarity inference from multiple perspectives, whereas most methods simply analyze a single angle. To our knowledge, it is the first study to consider vector-based similarity representations and learning from multiple perspectives that are complementary. It enables more comprehensive learning of cross-modal correspondences.

## 3 METHODOLOGY

As shown in Figure 2, we first construct the semantic knowledge graphs. The KGID modules are then developed to learn fine-grained modal representations. Subsequently, vector-based similarity representations are learned from two perspectives: distance or direction similarity, to comprehensively explore multi-level correspondences. Finally, the SRGR module, which promotes the interaction between

global and local alignments by constructing and adaptively reconstructing similarity relation graphs.

## 3.1 SEMANTIC KNOWLEDGE GRAPH CONSTRUCTION

### 3.1.1 Visual Semantic Knowledge Graph

Given a raw image $I$, we use the Faster-RCNN [Krishna et al., 2017], which is pre-trained on Visual Genome, to detect $n$ ($n = 36$) prominent regions. Then, we add a fully connected layer to transform them into D-dimensional vectors to obtain region representations $V = \{v_1, v_2, \ldots, v_n\}$.

Formally, we aim to create an undirected, fully connected visual semantic knowledge graph $G_v = \{V_v, E_v\}$ for each image, with the detected regions set as nodes, and the edge denoted as a matrix $E_v$. On the one hand, there are spatial dependencies between regions. For example, "people on the court" and "people outside the court" reflect the spatial location relationship between "people" and "court". Thus, we follow [Norcliffe-Brown et al., 2018] in modeling spatial dependencies between regions using polar coordinates and representing them as a spatial dependence matrix $P_v$. On the other hand, there are also potential semantic correlations between regions. For example, the semantic information "playing tennis" is formed by associating the three regions: "women", "racket" and "tennis". Therefore, we calculate the semantic correlation matrix $r^v$ between regions:

$$r_{ij}^v = \frac{\exp\left(\lambda v_i^\mathsf{T} v_j\right)}{\sum_{j=0}^{v} \exp\left(\lambda v_i^\mathsf{T} v_j\right)}, \qquad (1)$$

where $\lambda$ is the scale factor. $r_{ij}^v$ denotes the correlation between the $i$-th region and the $j$-th region. The visual semantic knowledge graph is made up of the spatial interdependence and the semantic correlations between regions. We calculate the Hadamard product of $r^v$ and $P_v$, then apply column L2-normalization to obtain the edge matrix $E_v$.

$$E_v = \|r^v \odot P_t\|_2, \qquad (2)$$

where $\|\cdot\|_2$ denotes column L2-normalization.

### 3.1.2 Textual Semantic Knowledge Graph

For a text $T$ comprising $m$ words, we first represent each word as a continuous embedding vector. Then the word vectors are embedded into a bi-directional GRU [Cho et al., 2014]. Finally, textual word feature representation is obtained, denoted as $T = \{t_1, t_2, \ldots, t_m\}$.

To construct a textual semantic knowledge graph $G_t = \{V_t, E_t\}$, we set words as nodes, which are semantically related to each other. To obtain the syntactic dependency matrix $P_t$, we first utilize Stanford CoreNLP to find syntactic

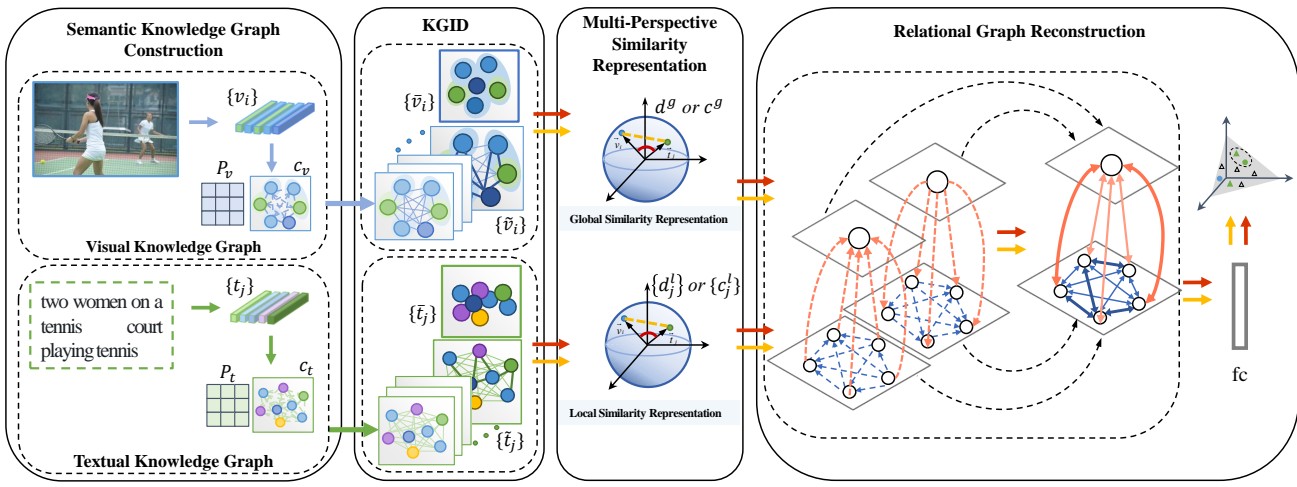

Figure 2: An overview of our MPSM. It is made up of four modules: (1) Semantic knowledge graph construction: extracting features and constructing semantic knowledge graphs; (2) KGID module: nvestigating semantically relevant features; (3) Multi-Perspective Similarity Representation: learning vector-based similarity representation from distance (yellow path) or direction (red path) similarity; (4) SRGR module: enhancing relevant similarities.

dependencies, and add self-loops (the matrix diagonal is 1). We compute the correlation $r^t$ between words. Similarly, the $E_t$ is then obtained by performing a column l2-normalizatio operation on the Hadamard product of $r^t$ and $P_t$ :

$$r_{ij}^t = \frac{\exp\left(\lambda t_i^\top t_j\right)}{\sum_{j=0}^m \exp\left(\lambda t_i^\top t_j\right)}, \quad (3)$$

$$E_t = \left\| r^t \odot P_t \right\|_2, \quad (4)$$

## 3.2 KNOWLEDGE GRAPH ITERATIVE DISSEMINATION

Considering that both images and texts are based on the KGID module for knowledge propagation, we first depict the knowledge propagation process on the visual semantic knowledge graph in detail, and then roughly on the textual.

### 3.2.1 Visual KGID

Given a visual semantic knowledge graph $G_v = \{V_v, E_v\}$, its node representation denoted as $V$. We use the associated edges to associate each node with other nodes and propagate knowledge, resulting in a new visual semantic association feature map. Then, the softmax function is used to learn each region's knowledge weight coefficients and update the nodes' own features by element multiplication. Therefore, we obtain the the "knowledgeable" local representation by

$$V_k^{(l)} = \rho\left(A^{(l)} V W^{(l)}\right) \odot V, \quad (5)$$

where $A^0 = E_v$, $W^{(l)} \in \mathbb{R}^{d \times d}$, $\rho$ is the softmax function.

In order to aggregate and enhance semantic correlations, we design an adaptive gate mechanism with fusion and

reconstruction procedures in the process of knowledge propagation and reasoning. We obtain gating mask by combining $V_k^{(l)}$ and $V$. Then the gating mask are utilized to control the flow of information between $V_k^{(l)}$ and $V$. Therefore the image feature representation is dynamically updated by

$$g^{(l)} = \sigma\left(\left(V_k^{(l)} \odot V\right) + V\right), \quad (6)$$

$$V_{kg}^{(l)} = g^{(l)} \odot V_k^{(l)} + (1 - g^l) \odot V, \quad (7)$$

where $\sigma$ is the sigmoid function, $\odot$ is the Hadamard product.

Finally, we perform aggregated inference followed by shortcut connection to enhance the local feature representation:

$$V^{(l+1)} = \text{ReLu}\left(\left(W_{kg}^{(l)} V_{kg}^{(l)} + b^{(l)}\right) + V\right), \quad (8)$$

where $W_{kg}^{(l)} \in \mathbb{R}^{d \times d}$. Moreover, we follow [Kuang et al., 2019] to update the edge $A^{(l+1)}$ using the affinity of new nodes, i.e., by adaptively update the semantic knowledge through the affinity between regions:

$$A^{(l+1)} = \text{softmax}_j\left(\lambda(W_O^{(l+1)} V_i^{(l+1)}) \times (W_I^{(l+1)} V_j^{(l+1)})\right), \quad (9)$$

where $W_I^{(l+1)} \in \mathbb{R}^{d \times d}$ and $W_O^{(l+1)} \in \mathbb{R}^{d \times d}$ are linear transformations of incoming and outgoing nodes, respectively. $A^{(l+1)}$ means that if two regions are highly correlated, edges with higher scores will connect the nodes.

We iteratively infer, update, and aggregate the visual semantic knowledge graph in $N$ steps, and use the output of the last step as the local inference representation $\tilde{v}$ of the image.

To obtain the global feature, we perform a self-attentive operation on the local region features. Specifically, given the mean-pooled feature $v_m = \frac{1}{N}\sum_{i=1}^N \tilde{v}_i$,

with $v_m$ as the query, and $\tilde{v}$ as the key and value, we first calculate the attention weight distribution $a^v = \mathrm{softmax}\left(W_1^v\left((W_2^v\tilde{v}) \odot (W_3^v v_m)\right)\right)$ for all regions, where $W_1^v \in \mathbb{R}^{1 \times d}$, $W_2^v \in \mathbb{R}^{n \times d}$ and $W_3^v \in \mathbb{R}^{n \times d}$. Then, the image global feature is obtained by $\bar{v} = \frac{1}{N}\sum_{i=1}^{N} a_i^v \tilde{v}_i$.

### 3.2.2 Textual KGID

Similarly, given a textsual semantic knowledge graph $G_t = \{V_t, E_t\}$, its node representation is denoted as $T$, we perform an $N$-step iterative propagation and inference to mine fine-grained local features $\tilde{t}$. Subsequently, a self-attentive operation is performed to mine the global features $\bar{t}$.

### 3.3 MULTI-PERSPECTIVE SIMILARITY REPRESENTATION LEARNING

We learn local and global similarity representations by using distance or directional differences between vectors. It enables a more comprehensive understanding of similarity.

In order to enhance the visual representation, we use the words information from each sentence as cues to focus on all regions in each image. For each image, we first compute the cosine similarity between each region and each word to establish the relationship $R$. The softmax function is then used to calculate the attention weights. Finally, we construct the augmented representation of images associated with the $j$-th word by $v_j^t = \sum_{i=1}^{m} \mathrm{softmax}\left(R_{ij}\right)\tilde{v}_i$.

### 3.3.1 Distance Similarity

To learn the distance similarity between vectors, we first calculate the vector-based squared Euclidean distance between vectors $x \in \mathbb{R}^d$ and $y \in \mathbb{R}^d$ by $dis(x, y) = (x - y)^2$. Then, the distance similarity function denotes as

$$d(x, y, W_d) = \frac{W_d \cdot dis(x, y)}{\|W_d \cdot dis(x, y)\|_2}, \qquad (10)$$

where $W_d \in \mathbb{R}^{m \times d}$ is a learnable parameter matrix.

Using the Eq.(10), we compute the local similarity representation $d_j^l$ between $v_j^t$ and $\tilde{t}_j$, then calculate the global similarity representation $d^g$ between $\bar{v}$ and $\bar{t}$:

$$d_j^l = d\left(v_j^t, \tilde{t}_j, W_d^l\right), \qquad (11)$$
$$d^g = d\left(\bar{v}, \bar{t}, W_d^g\right), \qquad (12)$$

### 3.3.2 Direction Similarity

We consider the similarity representation learning based on the cosine distance between $x \in \mathbb{R}^d$ and $y \in \mathbb{R}^d$. Thus, the "direction" similarity representation is defined as

$$c(x, y, W_c) = \frac{W_c \, dir(x, y)}{\|W_c \, dir(x, y)\|_2}, \qquad (13)$$

where $dir(x, y) = (x \cdot y) / (\|x\| \cdot \|y\|)$.

We calculate the local similarity representation $c_j^l$ between feature $v_j^t$ and $\tilde{t}_j$ with Eq.(13), and calculate the global similarity representation $c^g$ between $\bar{v}$ and $\bar{t}$:

$$c_j^l = c\left(v_j^t, \tilde{t}_j, W_c^l\right), \qquad (14)$$
$$c^g = c\left(\bar{v}, \bar{t}, W_c^g\right), \qquad (15)$$

### 3.4 SIMILARITY-RELATIONAL GRAPH RECONSTRUCTION

#### 3.4.1 Relational Graph Building

Formally, we construct a directed relational weighted graph of similarity representations. Specifically, we denote all "distance" (or "direction") similarity representations as graph nodes $\mathcal{N} = \left\{s^g, s_1^l, s_2^l, ..., s_j^l\right\}$, where $s^g$ denotes $d^g$ (or $c^g$), $s_j^l$ denotes $d_j^l$ (or $c_j^l$). For any node, the relationship between nodes is extracted from node $\mathbf{s}_a$ to node $\mathbf{s}_b$ and is defined as a variable edge weight by

$$E_{(s_a, s_b)}\left(W_{out}, W_{in}\right) = \frac{\sigma(BN(W_{out}\mathbf{s}_a) \oplus BN(W_{in}\mathbf{s}_b))}{\sum_{s_i \in \mathcal{N}} \sigma(BN(W_{out}\mathbf{s}_a) \oplus BN(W_{in}\mathbf{s}_b))}$$
$$(16)$$

where $W_{out} \in \mathbb{R}^{m \times 1}$ and $W_{in} \in \mathbb{R}^{1 \times m}$ are the linear transformations of outgoing and incoming nodes, the "$\oplus$" indicates concatenation. $R\left(s_a, s_b\right) = BN\left(W_{out}s_a\right) \oplus \left(BN\left(W_{in}s_b\right)\right)$ is the trend score of the node-node relationship, and the edge weights $E_{(s_a, s_b)}\left(W_{out}, W_{in}\right)$ can be calculated by using sigmoid function $\sigma$. Note that $s_a \rightarrow s_b$ differs from $s_a \leftarrow s_b$, i.e. the edges are directed.

#### 3.4.2 Relational Graph Reconstruction

We perform a series of processes, such as propagation of similarity relations and gate mechanisms, to achieve the interaction of similar information and the reconstruction of the similarity-relational graph. Since the relational edges are directed, we take the outgoing and incoming inference, respectively, to implement bi-SRGR:

$$\begin{aligned}\overrightarrow{\tilde{s}}_a &= \sum_{\mathbf{s}_b \in \mathcal{N}} \overrightarrow{E_{(s_a, s_b)}}\left(W_{out}, W_{in}\right) \cdot \mathbf{s}_b, \\ \overleftarrow{\tilde{s}}_a &= \sum_{\mathbf{s}_b \in \mathcal{N}} \overleftarrow{E_{(s_a, s_b)}}\left(W_{in}, W_{out}\right) \cdot \mathbf{s}_b, \end{aligned} \qquad (17)$$

where $W_{out} \in \mathbb{R}^{m \times m}$ and $W_{in} \in \mathbb{R}^{m \times m}$. The edge weights of $s_a$ output and input are denoted by $\overrightarrow{E_{(s_a, s_b)}}\left(W_{out}, W_{in}\right)$ and $\overleftarrow{E_{(s_a, s_b)}}\left(W_{in}, W_{out}\right)$, respectively. $\overrightarrow{\tilde{s}}_a$ and $\overleftarrow{\tilde{s}}_a$ denote the results of propagating all similarity information from node $s_a$ outgoing and incoming, respectively, both of which contain the same node $s_a$.

Furthermore, to improve the quality of dynamic decision-making, we propose a conditional selection strategy to adaptively filter node information and suppress unnecessary information. Specifically, $\overrightarrow{\tilde{s}}_a$ and $\overleftarrow{\tilde{s}}_a$ are first concatenated,

followed by a fully connected layer and a sigmoid function to obtain the conditional mask,

$$\tilde{g} = \sigma \left( \widetilde{W} \left( \overrightarrow{\tilde{s}_a} \oplus \overleftarrow{\tilde{s}_a} \right) + \tilde{b} \right). \qquad (18)$$

Then, we use the generated conditional control masks to control the information flow of the original $\overrightarrow{\tilde{s}_a}$ and $\overleftarrow{\tilde{s}_a}$, followed by a shortcut connection to achieve an adaptively filtered and enhanced similarity representation, thus the reconstructed $\overrightarrow{\tilde{s}_a^*}$ and $\overleftarrow{\tilde{s}_a^*}$ can be achieved by

$$\begin{aligned} \overrightarrow{s_a^*} &= W_1^* \left( \tilde{g} \odot \overrightarrow{\tilde{s}_a} \right) + \overrightarrow{\tilde{s}_a}, \\ \overleftarrow{s_a^*} &= W_2^* \left( \tilde{g} \odot \overleftarrow{\tilde{s}_a} \right) + \overleftarrow{\tilde{s}_a}. \end{aligned} \qquad (19)$$

Furthermore, we aggregate $\overrightarrow{\tilde{s}_a^*}$ and $\overleftarrow{\tilde{s}_a^*}$, followed by a fully connected layer, n, which is formulated as,

$$s_a^* = W^* \left( \overrightarrow{s_a^*} + \overleftarrow{s_a^*} \right) + b^*, \qquad (20)$$

where $W^* \in \mathbb{R}^{m \times m}$. Finally, we feed $s_a^*$ into a fully connected layer to predict the final similarity score.

## 3.5 TRAINING OBJECTIVES AND INFERENCE STRATEGIES

We employ bidirectional triplet ranking loss as the objective function. Given a representation of the matched image-text pair $(v, t)$, its corresponding negative pairs are denoted as $(t, v^-)$ and $(v, t^-)$. We compute the loss with

$$\begin{aligned} \mathcal{L}_{\text{dis}}(v, t) = \sum_{(v,t)} &\{ \max \left[ 0, \gamma - \mathcal{S}_{dis}(v, t) + \mathcal{S}_{dis}(v, t^-) \right] \\ &+ \max \left[ 0, \gamma - \mathcal{S}_{dis}(v, t) + \mathcal{S}_{dis}(v^-, t) \right] \}. \end{aligned} \qquad (21)$$

where $\mathcal{S}_{dis}(v, t)$ is the similarity prediction function based on the "distance" similarity representation. Similarly, we define the ranking loss of MPSM(dir) as $\mathcal{L}_{\text{dir}}$.

In this paper, we use the proposed "distance" and "direction" similarity representations to investigate two training and inference strategies: joint training and independent training. For joint training, We combine $\mathcal{L}_{\text{dis}}$ and $\mathcal{L}_{\text{dir}}$ to train our MPSM model, i.e., we combine the "distance" and the "direction" similarity representation for training. For independent training, we train two single model, MPSM (dis) based on the "distance" similarity representation and MPSM (dir) based on the "direction" similarity representation. Then, in the inference phase, we average the similarities predicted by the MPSM (dis) and the MPSM (dir) for retrieval evaluation.

## 4 EXPERIMENTS

### 4.1 DATASETS AND IMPLEMENTATION DETAILS

The Flickr30K dataset [Plummer et al., 2015] and MS-COCO dataset [Lin et al., 2014] (1K and 5K test set) were

used for validating the effectiveness our proposed method. We utilize the typical Recall@K (K=1, 5, 10) as the performance evaluation metric. We trained our model with Adam optimizer with 30 epochs on the Flickr30K dataset and 20 epochs on the MS-COCO dataset. The dimensionality of the similarity representation to 256, and the other parameters are set to: $l = 3, \gamma = 0.2, \lambda = 9$.

### 4.2 QUANTITATIVE RESULTS

We compare the proposed MPSM with several state-of-the-art baselines. Note that the majority of these models are ensemble models. Therefore, we provide two versions of MPSM: MPSM (dis) and MPSM (dir) that based on the "distance" and the "direction" similarity representation, respectively. Then, we integrate them by averaging their similarity scores, and denotes as MPSM*.

#### 4.2.1 Results on Flickr30K Dataset

The quantitative results on the Flickr30K dataset are shown in Table 1, and it can be observed that our MPSM model outperforms the state-of-the-art in most assessment measures. Compared with GSMN, our method outperforms it in all metrics. Unlike GSMN, our approach propagates and aggregates semantic knowledge, rather than performing image-text matching directly. Furthermore, we simulate the interaction of global and local alignments, which obtains more comprehensive cross-modal correlations. Improvements show that propagating semantic information to learn fine-grained intra-modal correlations and incorporating them into cross-modal similarity learning improves matching performance significantly. Our proposed method outperforms other models that use the same word feature learning method (i.e., bi-GRU). Compared to CAMERA, our method achieves relative R@1 gains of 2.2% and 1.2% for I2T and T2I matching, respectively. However, our method reduces the relative R@5 and R@10 to 1.4% and 1.6% for T2I matching, respectively. This could be because CAMERA employs a pre-trained BERT. BERT learns feature representations of words based on a massive corpus, with powerful language representation and sentence processing capabilities.

When compared to SGRAF, a multi-level alignment learning method that also employs vector-based similarity representation, our method achieves relative R@1 gains of 2.4% and 3% for I2T and T2I matching, respectively. Unlike SGRAF, we model the vector-based similarity representation from two perspectives: distance and direction. Furthermore, our SRGR module makes the visual-semantic correspondence more fine-grained. Moreover, our KGID module provides rich semantic information within modalities. The advancements demonstrate that learning similarity from multiple perspectives, can help with cross-modal alignment.

It's worth mentioning that compared with MPSM (dis) and

Table 1: Results on Flickr30K and MSCOCO. * indicates to the ensemble result. The best result is marked in bold.

| Method | Flickr30K dataset | | | | | | MSCOCO 1K Test Set | | | | | | MSCOCO 5K Test Set | | | | | |
|---|---|---|---|---|---|---|---|---|---|---|---|---|---|---|---|---|---|---|
| | Image to Text | | | Text to Image | | | Image to Text | | | Text to Image | | | Image to Text | | | Text to Image | | |
| | R@1 | R@5 | R@10 | R@1 | R@5 | R@10 | R@1 | R@5 | R@10 | R@1 | R@5 | R@10 | R@1 | R@5 | R@10 | R@1 | R@5 | R@10 |
| VSE++ [Faghri et al., 2018] | 52.9 | 79.1 | 87.2 | 39.6 | 69.6 | 79.5 | 64.6 | 90.0 | 95.7 | 52.0 | 84.3 | 92.0 | 41.3 | 71.1 | 81.2 | 30.3 | 59.4 | 72.4 |
| MTFN [Wang et al., 2019a] | 65.3 | 88.3 | 93.3 | 52.0 | 80.1 | 86.1 | 74.3 | 94.9 | 97.9 | 60.1 | 89.1 | 95.0 | 48.3 | 77.6 | 87.3 | 35.9 | 66.1 | 76.1 |
| SCAN* [Lee et al., 2018] | 67.4 | 90.3 | 95.8 | 48.6 | 77.7 | 85.2 | 72.7 | 94.8 | 98.4 | 58.8 | 88.4 | 94.8 | 50.4 | 82.2 | 90.0 | 38.6 | 69.3 | 80.4 |
| VSRN* [Li et al., 2019] | 71.3 | 90.6 | 96.0 | 54.7 | 81.8 | 88.2 | 76.2 | 94.8 | 98.2 | 62.8 | 89.7 | 95.1 | 53.0 | 81.1 | 89.4 | 40.5 | 70.6 | 81.1 |
| IMRAM* [Chen et al., 2020] | 74.1 | 93.0 | 96.6 | 53.9 | 79.4 | 87.2 | 76.7 | 95.6 | 98.5 | 61.7 | 89.1 | 95.0 | 53.7 | 83.2 | 91.0 | 39.7 | 69.1 | 79.8 |
| MMCA [Wei et al., 2020] | 74.2 | 92.8 | 96.4 | 54.8 | 81.4 | 87.8 | 74.8 | 95.6 | 97.7 | 61.6 | 89.8 | 95.2 | 54.0 | 82.5 | 90.7 | 38.7 | 69.7 | 80.8 |
| GSMN* [Liu et al., 2020] | 76.4 | 94.3 | 97.3 | 57.4 | 82.3 | 89.0 | 78.4 | 96.4 | 98.6 | 63.3 | 90.1 | 95.7 | - | - | - | - | - | - |
| CAMERA* [Qu et al., 2020] | 78.0 | **95.1** | 97.9 | 60.3 | **85.9** | **91.7** | 77.5 | 96.3 | 98.8 | 63.4 | 90.9 | 95.8 | 55.1 | 82.9 | 91.2 | 40.5 | 71.7 | 82.5 |
| SMFEA [Ge et al., 2021] | 73.7 | 92.5 | 96.1 | 54.7 | 82.1 | 88.4 | 75.1 | 95.4 | 98.3 | 62.5 | 90.1 | 96.2 | 54.2 | - | 89.9 | 41.9 | - | 83.7 |
| CASC [Xu et al., 2020] | 68.5 | 90.6 | 95.9 | 50.2 | 78.3 | 86.3 | 72.3 | 96.0 | 99.0 | 58.9 | 89.8 | 96.0 | 47.2 | 78.3 | 87.4 | 34.7 | 64.8 | 76.8 |
| SHAN* [Ji et al., 2021] | 74.6 | 93.5 | 96.9 | 55.3 | 81.3 | 88.4 | 76.8 | 96.3 | 98.7 | 62.6 | 89.6 | 95.8 | - | - | - | - | - | - |
| SGRAF* [Diao et al., 2021] | 77.8 | 94.1 | 97.4 | 58.5 | 83.0 | 88.8 | 79.6 | 96.2 | 98.5 | 63.2 | 90.7 | 96.1 | 57.8 | - | 91.6 | 41.9 | - | 81.3 |
| **MPSM (dis)** | 77.5 | 94.0 | 97.0 | 58.7 | 83.6 | 89.1 | 78.4 | 96.0 | 98.5 | 63.1 | 90.0 | 95.6 | 58.1 | 84.3 | 91.4 | 41.5 | 70.9 | 81.4 |
| **MPSM (dir)** | 76.8 | 94.3 | 97.0 | 57.3 | 82.9 | 88.9 | 78.4 | 96.3 | 98.8 | 63.5 | 90.4 | 95.8 | 57.5 | 84.4 | 91.7 | 41.7 | 71.2 | 81.5 |
| **MPSM*** | **80.2** | 94.9 | **98.0** | **61.5** | 84.5 | 90.1 | **80.9** | **96.5** | **99.0** | **65.0** | **91.1** | **96.1** | **60.3** | **86.1** | **92.5** | **43.5** | **72.8** | **82.8** |

MPSM (dir), MPSM* has increased by 2.7% and 3.4% in I2T retrieval, has increased by 2.8% and 4.2% in T2I retrieval relative to R@1, respectively. This demonstrates that the MPSM(dis) and MPSM(dir) models can complement and enhance each other, allowing for a more comprehensive exploration of the correspondence between modalities. Furthermore, our single model's retrieval performance is very competitive, demonstrating the effectiveness of our method.

### 4.2.2 Results on MSCOCO Dataset

Table 1 shows quantitative results for the larger and more complex dataset MSCOCO (1K and 5K test sets). Our MPSM surpasses existing approaches in all metrics in the 1K testset. Compared with GSMN, our method outperforms it in all metrics. Compared to SGRAF, on I2T and T2I retrieval, our MPSM improves by 1.3% and 1.8%, respectively. The gain in our method's performance over R@5 and R@10 is not as large as it is for R@1, which could be owing to the presence of more interference sources in a big target set for a particular query. Our MPSM maintains its superiority in the 5K testset. Our model outperforms SGRAF by 2.5% and 1.6 % in I2T and T2I retrieval, respectively.

### 4.3 ABLATION STUDIES AND ANALYSIS

#### 4.3.1 Impact of Different Network Structures

We compare MPSM (dis) and the integrated model MPSM with four other models (based on "distance" similarity representation). (1) w/o KGID denotes the removal of the whole KGID module from the model; (2) w/o V-KGID denotes the absence of the visual KGID module; (3) w/o T-KGID denotes the absence of the textual KGID module; and (4) w/o SRGR denotes the absence of the SRGR module.

As shown in Table 2, both MPSM (dis) and MPSM outperform these four types of models. Specifically, when we

remove the KGID module, the model performance suffers, which justifies the usage of modality-specific semantic information to investigate fine-grained semantic correlations and modal representations. Noting that the performance of w/o V-KGID and w/o T-KGID is better than that of w/o KGID, owing to the inclusion of semantic information from text or image, which can help with cross-modal correspondence investigation. The performance of MPSM (dis) is superior to that of w/o SRGR, demonstrating the effectiveness of the SRGR module. It also demonstrates that in-depth exploration of cross-modal similarity relation can facilitate aggregation and enhance similarity for more accurate matching.

Table 2: Impact of different structures on Flickr30K.

| Modal | Image to Text | | | Text to Image | | |
|---|---|---|---|---|---|---|
| | R@1 | R@5 | R@10 | R@1 | R@5 | R@10 |
| w/o KGID | 75.8 | 93.2 | 96.3 | 57.0 | 81.5 | 88.4 |
| w/o T-KGID | 76.2 | 93.9 | 96.9 | 56.6 | 82.5 | 88.1 |
| w/o V-KGID | 75.2 | 94.0 | 96.6 | 56.8 | 82.7 | 88.7 |
| w/o SRGR | 75.2 | 93.7 | 97.1 | 57.7 | 82.1 | 89.0 |
| MPSM (dis) | 77.5 | 94.0 | 97.0 | 58.7 | 83.6 | 89.1 |
| MPSM* | 80.2 | 94.9 | 98.0 | 61.5 | 84.5 | 90.1 |

#### 4.3.2 Impact of Different KGID Layers

We researched the impact layers of KGID modules on performance, gradually increasing the number of KGID layers from 0 to 4 for training and evaluation. As can be seen in Table 3, increasing the KGID improves performance. The model performs best when the number of KGID layers is increased to 3, demonstrating that iteratively propagating semantic knowledge is effective in boosting performance. This is because, during knowledge dissemination, the KGID module may integrate nodes' domain information and build connections with related nodes. The performance of KGID degrades as the number of layers increases to 4. This could be due to the fact that as the network grows deeper, the noise

level rises in tandem with the number of connected nodes, interfering with the learnt correspondence. As a result, we finally set the KGID module to 3 layers.

Table 3: Impact of different KGID layers on Flickr30K.

| Modal | Image to Text | | | Text to Image | | |
|---|---|---|---|---|---|---|
| | R@1 | R@5 | R@10 | R@1 | R@5 | R@10 |
| w/o KGID | 75.8 | 93.2 | 96.3 | 57.0 | 81.5 | 88.4 |
| 1KGID | 75.4 | 93.7 | 96.5 | 57.6 | 82.9 | 88.8 |
| 2KGID | 75.9 | 93.8 | 96.5 | 57.8 | 82.9 | 88.9 |
| 3KGID | **77.5** | **94.0** | **97.0** | **58.7** | **83.6** | **89.1** |
| 4KGID | 76.4 | 93.6 | 96.9 | 56.9 | 82.5 | 88.3 |

### 4.3.3 Impact of Different Alignment Strategies

We investigated three alignments: (1) global alignment learning strategy, which implies that only global alignment is used in the model; (2) local alignment learning strategy, implies that only local alignment is used; and (3) multi-level alignment learning strategy, which indicates that global and

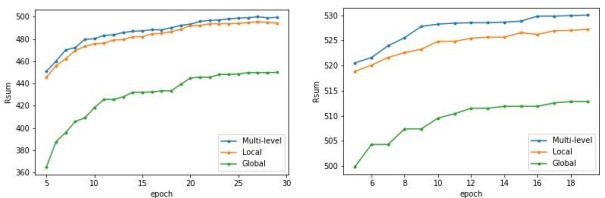

(a) Rsum results on Flickr30K   (b) Rsum results on MSCOCO

Figure 3: Comparison of Rsum results on Flickr30K and MSCOCO 1K test set with different alignment strategies.

local alignment are combined. As shown in Figures 3, the model's performance decreases dramatically when only the global correspondence is considered, without taking into account the relationship between local and global. Moreover, when compared to local alignment learning, the multilevel alignment learning strategy achieves superior performance. It demonstrates that global and local alignments can complement each other's semantic information to achieve more accurate matching by interacting with "part" and "whole".

### 4.3.4 Impact of Training Strategies

We design two different training strategies, "joint training" and "independent training with integration", and compare them. From Table 4, we can see that the "independent training and integration" strategy achieves superior performance than the "joint training" strategy. On the one hand, the training with individual learners tends to cause underfitting or overfitting, resulting in insufficient generalization ability of the joint training strategy. Instead, we train MPSM (dis) and MPSM (dir) separately, and integrate them by calculating their means to complement each other, resulting in an

ensemble modal with superior generalization performance. On the other hand, the "distance" similarity representation focuses on measuring the magnitude of similarity while ignoring directional differences between images and text; the "direction" similarity representation distinguishes the difference between vectors more from direction than numerical value, and thus fails to quantify the image-text correspondence finely. However, they are complementary. Thus, the ensemble model MPSM is based on "distance" and "direction" similarity representations, which can facilitate the exploration of fine-grained cross-modal correspondences.

Table 4: Impact of different KGID layers.

| Modal | distance | direction | Joint | Split | Image to Text | | Text to Image | |
|---|---|---|---|---|---|---|---|---|
| | | | | | R@1 | R@10 | R@1 | R@10 |
| Flickr30K | ✓ | | | | 77.5 | 97.0 | 58.7 | 89.1 |
| | | ✓ | | | 77.0 | 97.0 | 57.3 | 88.9 |
| | ✓ | ✓ | ✓ | | 77.1 | 97.2 | 59.3 | 88.7 |
| | ✓ | ✓ | | ✓ | **80.2** | **98.0** | **61.5** | **90.1** |
| MSCOCO 1K | ✓ | | | | 78.4 | 98.5 | 63.1 | 95.6 |
| | | ✓ | | | 78.4 | 98.8 | 63.5 | 95.8 |
| | ✓ | ✓ | ✓ | | 79.5 | **99.0** | 63.6 | 95.9 |
| | ✓ | ✓ | | ✓ | **80.9** | **99.0** | **65.0** | **96.1** |

## 4.4 QUALITATIVE RESULTS AND ANALYSIS

Furthermore, we show the qualitative results of I2T and T2I retrieval on the Flickr30K in Figure 4. For the I2T retrieval in (a), we show the top-5 retrieved sentences based on our predicted similarity score ranking. Our model can retrieve

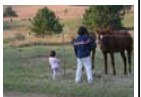

1. Two children are in a grassy area near two horses. ✓
2. 2 kids talk to the horses. ✓
3. Two children looking at horses through a small fence. ✓
4. Two children feeding horses through a fence. ✓
5. Two children pet horses in a field. ✓

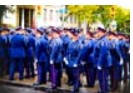

1. A group of men in blue uniforms are standing together. ✓
2. Men dressed up in blue uniforms standing in formation. ✓
3. Military men in blue and red suits stand in the street. ✓
4. A crowd of guards standing on a sidewalk. ✓
5. Men in uniform are standing on a street. ✗

(a) image-to-text matching

Text Query1:Two people bicycle on a path separated by small mountains.

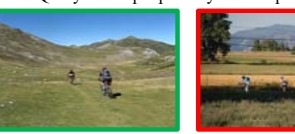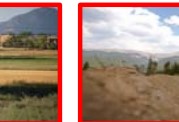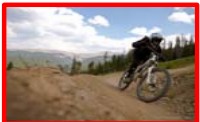

Text Query2: Two dogs are biting each other in the woods.

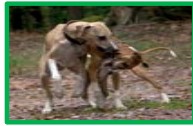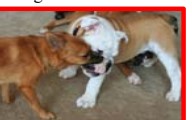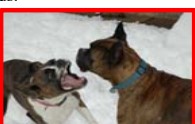

(b) text-to-image matching

Figure 4: Visualization of image-text retrieval on Flickr30K.

almost all sentences that match the query image; even the incorrect instances have some similarity. For example, the "men" region corresponds to the word "men", and the men are wearing uniforms which corresponds to the phrase "men in uniform". Thus, the semantics between the matched sentence "5" and the query image are almost identical. This is due to the KGID module, which investigates fine-grained correlations between fragments. In addition, MPSM considers similarity from multiple perspectives and SRGR module explore more comprehensive similarity and more precise matching. As for the T2I retrieval, we show the top-3 retrieved images and mark the correct results with green boxes. The top-1 image is the ground-truth, and all other results are close to the sentence's semantics. These results demonstrate our model's ability to perform finer-grained matching.

# 5 CONCLUSIONS

In this paper, we propose a fine-grained matching with Multi-Perspective Similarity Modeling (MPSM) Network for cross-modal retrieval. Specifically, we develop a knowledge graph iterative dissemination module that iteratively propagates semantic knowledge to capture fine-grained intra-modal correlations and modal representations. Then, from multiple perspectives, we learn vector-based similarity representations to adequately learn multi-level correspondences. Further, we designed a relationship graph reconstruction module that focuses on aggregating and improving the similarity between similar modalities to be able to obtain more accurate matches. Experiments on both datasets show that our network is superior.

**Acknowledgements**

This work is supported by National Natural Science Foundation of China (Nos. 61966004, 61866004), Guangxi Natural Science Foundation (No. 2019GXNSFDA245018), Innovation Project of Guangxi Graduate Education (YCSW2022155), Guangxi "Bagui Scholar" Teams for Innovation and Research Project, Guangxi Talent Highland Project of Big Data Intelligence and Application, and Guangxi Collaborative Innovation Center of Multi-source Information Integration and Intelligent Processing. (Corresponding author: Zhixin Li.)

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
