# OpenReview forum: "Fine-Grained Matching with Multi-Perspective Similarity Modeling for Cross-Modal Retrieval"
_auai.org/UAI/2022/Conference — UAI 2022 Poster_

### Official Review · Reviewer_f23R · 2022-03-22

**Q2(1) Originality/Novelty:** 2
**Q2(2) Significance/Impact:** 2
**Q2(3) Correctness/Technical Quality:** 2
**Q2(6) Clarity Of Writing:** 2
**Q6 Overall Score:** 5
**Q8 Confidence In Your Score:** 3

**Q1 Summary And Contributions:**

This paper proposes a MPSM network for cross-model retrieval. This network consists of two modules. KGID module is designed to broadcast global knowledge and RGR module is developed to enhance cross-modal correspondence. Experiments show that our model outperforms state-of-the-art methods.

**Q2 Assessment Of The Paper:**

More detailed information regarding each of these aspects is given below:

**Q2(4) Quality Of Experiments (Optional):**

2: Fair: The experimental evaluation is weak: important baselines are missing, or the results do not adequately support the main claims.

**Q2(5) Reproducibility:**

2: Fair: Key resources (e.g., proofs, code, data) are unavailable but key details (e.g., proof sketches, experimental setup) are sufficiently well-described for an expert to confidently reproduce the main results.

**Q3 Main Strengths:**

This paper contributes new ideas and this idea seems works according to authors experiments. I think proposed fine-grained matching will benefit cross-modal retrieval.

I am not an expert on this topic and educationally guess that the technique in this paper is OK.

The experiments in this paper are sufficient. Quantitative results on Flickr30K and MSCOCO are reported. Ablation studies disclose the effect of the two modules.

This paper is well written. I can easily follow.


**Q4 Main Weakness:**

Section 3 consists of 5 sub sections and each subsection is comprised by server points. It is hard to understand so many technique points and the title of each subsection is not very friend for readers to understand the relationship of each points.

**Q5 Detailed Comments To The Authors:**

I think that section 3 should be reorganized.

**Q7 Justification For Your Score:**

Basically, this paper is well written and the technique seems sound and experiments valid the effect of proposed methods.

**Q9 Complying With Reviewing Instructions:**

1: Yes.

---

### Official Review · Reviewer_7A14 · 2022-04-12

**Q2(1) Originality/Novelty:** 2
**Q2(2) Significance/Impact:** 2
**Q2(3) Correctness/Technical Quality:** 3
**Q2(6) Clarity Of Writing:** 3
**Q6 Overall Score:** 5
**Q8 Confidence In Your Score:** 4

**Q1 Summary And Contributions:**

Motivations:
1 Consider both global and local alignment for fine-grained image-text retrieval ;
2 Existing methods fail to explore fine-grained intra-modal associations that provide richer semantic information.
Contributions:
1 The KGID module capture fine-grained local and global representations;
2 Similarity distance and similarity direction are considered simultaneously.
3 An RGR module is designed to construct and reconstruct the multilevel similarity diagram to enhance the similarity.

**Q2 Assessment Of The Paper:**

More detailed information regarding each of these aspects is given below:

**Q2(4) Quality Of Experiments (Optional):**

3: Good: The experimental evaluation is adequate, and the results convincingly support the main claims.

**Q2(5) Reproducibility:**

2: Fair: Key resources (e.g., proofs, code, data) are unavailable but key details (e.g., proof sketches, experimental setup) are sufficiently well-described for an expert to confidently reproduce the main results.

**Q3 Main Strengths:**

The most prominent advantage lies in the consideration of both similarity distance and direction for similarity representation and learning.

The proposed knowledge graph iterative propagation algorithm is used to explore fine-grained modal representation and the similarity is improved by constructing and adaptively reconstructing the similarity diagram.
The ABLATION STUDIES AND ANALYSIS is clear and thorough.


**Q4 Main Weakness:**

Compared with other methods, the advantages of the experimental results are reflected in the multi-perspective strategy, while the proposed KGID and RGR do not show advantages. Although the experiment in this paper contains many comparison methods, the two data sets are still a little insufficient.

**Q5 Detailed Comments To The Authors:**

Formula 18 has a clerical error.

**Q7 Justification For Your Score:**

The paper has clear and correct logical and easy-to-understand language, and presents novel algorithms. A detailed analysis of the experimental results is presented. However, there are only two experimental datasets, which is not enough.

The advantage of the experimental results over other methods is reflected in the multi-perspective strategy, which is not shown by the methods proposed in this paper (KGID and RGR).

The main strengths of the paper are given more weight.


**Q9 Complying With Reviewing Instructions:**

1: Yes.

---

### Official Review · Reviewer_4Fo1 · 2022-04-15

**Q2(1) Originality/Novelty:** 2
**Q2(2) Significance/Impact:** 2
**Q2(3) Correctness/Technical Quality:** 3
**Q2(6) Clarity Of Writing:** 4
**Q6 Overall Score:** 5
**Q8 Confidence In Your Score:** 1

**Q1 Summary And Contributions:**

The paper presents an approach for considering fine-grained correspondences in cross-modal retrieval, also taking into account multiple perspectives. The main contribution is a novel matching algorithm, Multi-Perspective Similarity Modeling (MPSM), based on a relation graph reconstruction module.

**Q2 Assessment Of The Paper:**

More detailed information regarding each of these aspects is given below:

**Q2(4) Quality Of Experiments (Optional):**

3: Good: The experimental evaluation is adequate, and the results convincingly support the main claims.

**Q2(5) Reproducibility:**

3: Good: Key resources (e.g., proofs, code, data) are available and key details (e.g., proofs, experimental setup) are sufficiently well-described for competent researchers to confidently reproduce the main results.

**Q3 Main Strengths:**

The topic addressed is interesting, the proposed approach is well described and extensively compared against several baselines. Also, an ablation study is performed.

**Q4 Main Weakness:**

The approach itself integrates well-know techniques. The difference with approaches in the baselines are not discussed to highlight the novelty of the approach.

**Q5 Detailed Comments To The Authors:**

The approach combines a number of well-know techniques to come up with an approach for fine-grained, cross-modal retrieval. In spite of addressing a relevant problem, the novelty of the approach is not sufficiently highlighted in the paper. Experimentation was done with two standard datasets and the performance of the approach was compared with several baselines. The approach outperformed the baselines in most cases, however, a detailed analysis of those cases in which is not the best performer would have enrich the discussion. Also, the differences of the baselines with the proposed approach need to be explained.


**Q7 Justification For Your Score:**

The approach is interesting although not high in novelty. The relation with baselines and discussion of results have room for improvement.

**Q9 Complying With Reviewing Instructions:**

1: Yes.

---

### Decision · Program_Chairs · 2022-05-15

**Decision:**

Accept (Poster)

**Comment:**

Meta Review: This paper develops an approach for fine-grained matching with multi-perspective similarity modeling for cross-modal retrieval. It contains two main novel modules. One is the knowledge graph iterative dissemination (KGID) module for iteratively broadcast global semantic knowledge and learn fined-grained modality representations. The relation graph reconstruction (RGR) module is developed to enhance cross-modal correspondence by adaptively reconstructing similarity relation graphs. The proposed model is well motivated and novel. Results also show that the model perform state of the art models. Overall this paper is a nice paper that UAI audience will be interested to hear about.